# Prenatal influenza vaccination and allergic and autoimmune diseases in childhood: A longitudinal, population-based linked cohort study

**Damien Foo**[1,2]*, **Mohinder Sarna**[1,2], **Gavin Pereira**[1,3,4], **Hannah C. Moore**[1,2], **Annette K. Regan**[1,2,5,6]

**1** Curtin School of Population Health, Curtin University, Perth, Western Australia, Australia, **2** Wesfarmers Centre of Vaccines and Infectious Diseases, Telethon Kids Institute, The University of Western Australia, Perth, Western Australia, Australia, **3** Centre for Fertility and Health, Norwegian Institute of Public Health, Oslo, Norway, **4** enAble Institute, Curtin University, Perth, Western Australia, Australia, **5** School of Nursing and Health Professions, University of San Francisco, San Francisco, California, United States of America, **6** Fielding School of Public Health, University of California Los Angeles, Los Angeles, California, United States of America

* Damien.Foo@postgrad.curtin.edu.au

**Data Availability Statement:** This study is part of a larger linked cohort study and was guided by the study protocol described elsewhere [22]. The

## Abstract

### Background

Few studies have evaluated the effect of maternal influenza vaccination on the development of allergic and autoimmune diseases in children beyond 6 months of age. We aimed to investigate the association between *in utero* exposure to seasonal inactivated influenza vaccine (IIV) and subsequent diagnosis of allergic and autoimmune diseases.

### Methods and findings

This longitudinal, population-based linked cohort study included 124,760 singleton, live-born children from 106,206 mothers in Western Australia (WA) born between April 2012 and July 2016, with up to 5 years of follow-up from birth. In our study cohort, 64,169 (51.4%) were male, 6,566 (5.3%) were Aboriginal and/or Torres Strait Islander children, and the mean age at the end of follow-up was 3.0 (standard deviation, 1.3) years. The exposure was receipt of seasonal IIV during pregnancy. The outcomes were diagnosis of an allergic or autoimmune disease, including asthma and anaphylaxis, identified from hospital and/or emergency department (ED) records. Inverse probability of treatment weights (IPTWs) accounted for baseline probability of vaccination by maternal age, Aboriginal and/or Torres Strait Islander status, socioeconomic status, body mass index, parity, medical conditions, pregnancy complications, prenatal smoking, and prenatal care. The models additionally adjusted for the Aboriginal and/or Torres Strait Islander status of the child. There were 14,396 (11.5%) maternally vaccinated children; 913 (6.3%) maternally vaccinated and 7,655 (6.9%) maternally unvaccinated children had a diagnosis of allergic or autoimmune disease, respectively. Overall, maternal influenza vaccination was not associated with

linked administrative data are de-identified and are not owned by the authors. Access to the data is approved by the Data Custodians and provided by the WA Data Linkage Branch within the WA Department of Health (https://www.datalinkage-wa.org.au/contact-us/). The use of the data is restricted to named researchers only. Programming code for data analysis is available in S1 Appendix.

**Funding:** This research was supported in part by funding received from the National Health and Medical Research Council (GNT1141510, PI: AKR, Investigators: MS, DF), Curtin University Graduate Research School, and the Wesfarmers Centre of Vaccines and Infectious Diseases at the Telethon Kids Institute. DF was supported by the Curtin University Graduate Research School and the Wesfarmers Centre of Vaccines and Infectious Diseases at the Telethon Kids Institute. AKR was supported by a National Health and Medical Research Council Fellowship (GNT1138425, PI: AKR). GP was supported by funding from the National Health and Medical Research Council Project and Investigator Grants (GNT1099655 and GNT1173991, PI: GP) and the Research Council of Norway through its Centres of Excellence funding scheme (#262700, PI: GP). HCM was supported by funding from a Telethon Kids Institute Emerging Research Leadership Fellowship and support from the Wesfarmers Centre of Vaccines and Infectious Diseases, Telethon Kids Institute. The funders had no role in the study design, data analysis, decision to publish, or preparation of the manuscript. The analyses, results, interpretation and conclusions reported in this manuscript are those of the authors and are independent from the funding sources.

**Competing interests:** We have read the journal's policy and the authors of this manuscript have the following competing interests: HCM has received consultancy payments from Merck Sharp & Dohme Australia (MSD) for participation in an Expert Input Forum and an investigator initiated grant from MSD. These are not related to the work under this submission. All authors have completed and submitted the ICMJE uniform disclosure form at www.icmje.org/coi_disclosure.pdf (available on request from the corresponding author).

**Abbreviations:** aHR, adjusted hazard ratio; CI, confidence interval; ED, emergency department; EDCC, Emergency Department Data Collection; HDMC, Hospital Morbidity Data Collection; HR, unadjusted hazard ratio; ICD-10-AM, International Statistical Classification of Diseases and Related Health Problems, Tenth Revision, Australian Modification; IIV, inactivated influenza vaccine;

diagnosis of an allergic or autoimmune disease (adjusted hazard ratio [aHR], 1.02; 95% confidence interval [CI], 0.95 to 1.09). In trimester-specific analyses, we identified a negative association between third trimester influenza vaccination and the diagnosis of asthma ($n = 40$; aHR, 0.70; 95% CI, 0.50 to 0.97) and anaphylaxis ($n = 36$; aHR, 0.67; 95% CI, 0.47 to 0.95). We did not capture outcomes diagnosed in a primary care setting; therefore, our findings are only generalizable to more severe events requiring hospitalization or presentation to the ED. Due to small cell sizes (i.e., <5), estimates could not be determined for all outcomes after stratification.

## Conclusions

In this study, we observed no association between *in utero* exposure to influenza vaccine and diagnosis of allergic or autoimmune diseases. Although we identified a negative association of asthma and anaphylaxis diagnosis when seasonal IIV was administered later in pregnancy, additional studies are needed to confirm this. Overall, our findings support the safety of seasonal inactivated influenza vaccine during pregnancy in relation to allergic and autoimmune diseases in early childhood and support the continuation of current global maternal vaccine programs and policies.

## Author summary

### Why was this study done?

- The World Health Organization recommends seasonal inactivated influenza vaccination for pregnant women to protect both the mother and her newborn from influenza infection.

- Few studies have evaluated the effect of maternal influenza vaccination on long-term health outcomes, including allergic and autoimmune diseases, in children exposed to influenza vaccines *in utero*.

### What did the researchers do and find?

- Using a large, population-based study of 124,760 children in Western Australia, we investigated the association between maternal influenza vaccination and allergic and autoimmune diseases.

- Prenatal administration of seasonal inactivated influenza vaccines was not associated with adverse allergic or autoimmune outcomes among children aged up to 5 years.

- We identified a negative association between prenatal influenza vaccination during the third trimester and asthma and anaphylaxis in children.

IPTW, inverse probability of treatment weight; MNS, Midwives Notification System; OR, odds ratio; RECORD, REporting of studies Conducted using Observational Routinely-collected Data; WA, Western Australia; WAAVD, WA Antenatal Vaccination Database.

## What do these findings mean?

- Our findings support the safety of seasonal inactivated influenza vaccine during pregnancy in relation to allergic and autoimmune diseases in early childhood. This information is reassuring and reinforces current global maternal vaccine programs and policies.

- While we identified a negative association for asthma and anaphylaxis when the vaccine was administered later in pregnancy, our findings are only generalizable to more severe events requiring hospitalization or presentation to the emergency department (ED).

- Additional research is needed to confirm and better elucidate our findings.

## Introduction

Influenza is a major respiratory infection that is linked with serious morbidity and mortality through seasonal epidemics each year, particularly among children aged <5 years [1,2], and while seasonal inactivated influenza vaccine (IIV) is the most effective preventative tool to protect against influenza infection [3], there are no current vaccines licensed for use for infants aged less than 6 months [2]. Previous studies have shown that vaccine-derived antibodies cross the placenta during pregnancy and offer passive immunity to infants during their first 6 months of life [2,4]. Prenatal administration of seasonal IIV is therefore recommended for pregnant women in many countries, including Australia, at any stage of pregnancy [2,4].

Substantial evidence supports the safety of maternal influenza vaccination with respect to health outcomes at birth, with no harmful association with preterm birth, small-for-gestational age, spontaneous abortion, stillbirth, low birth weight, congenital malformations, and fetal death [5–7]. However, according to a recent systematic review, few studies have assessed pediatric health outcomes beyond the first 6 months of life [8]. Previous clinical evidence has suggested that exposure to maternal vaccination *in utero* may "prime" the innate fetal immune system, resulting in a more activated and mature immunophenotype [9,10]. Further studies evaluating the impact of *in utero* exposure to seasonal influenza vaccines on the development of pediatric immune disorders are needed.

To our knowledge, only 3 studies have examined allergic or autoimmune outcomes among children >6 months of age, which highlights the novelty and the need for this study [11–13]. The aim of this study was to assess the risk of allergic and autoimmune diseases among maternally vaccinated and unvaccinated children with seasonal influenza vaccines.

## Methods

### Study cohort and design

This retrospective, population-based cohort study included all singleton, live-born children born in Western Australia (WA) between April 1, 2012 and July 1, 2016 and their mothers, as identified from birth registrations (**Fig 1**). Mother–child pairs were probabilistically linked with other population-based administrative health datasets by the WA Data Linkage Branch [14], including the Midwives Notification System (MNS) [15], the WA Antenatal Vaccination Database (WAAVD) [16], Hospital Morbidity Data Collection (HMDC) [17], Emergency Department Data Collection (EDDC) [18], and death registrations. The probabilistic linkage matches records from different sources using complex nonunique identifiers or field matching algorithms [19]. These algorithms

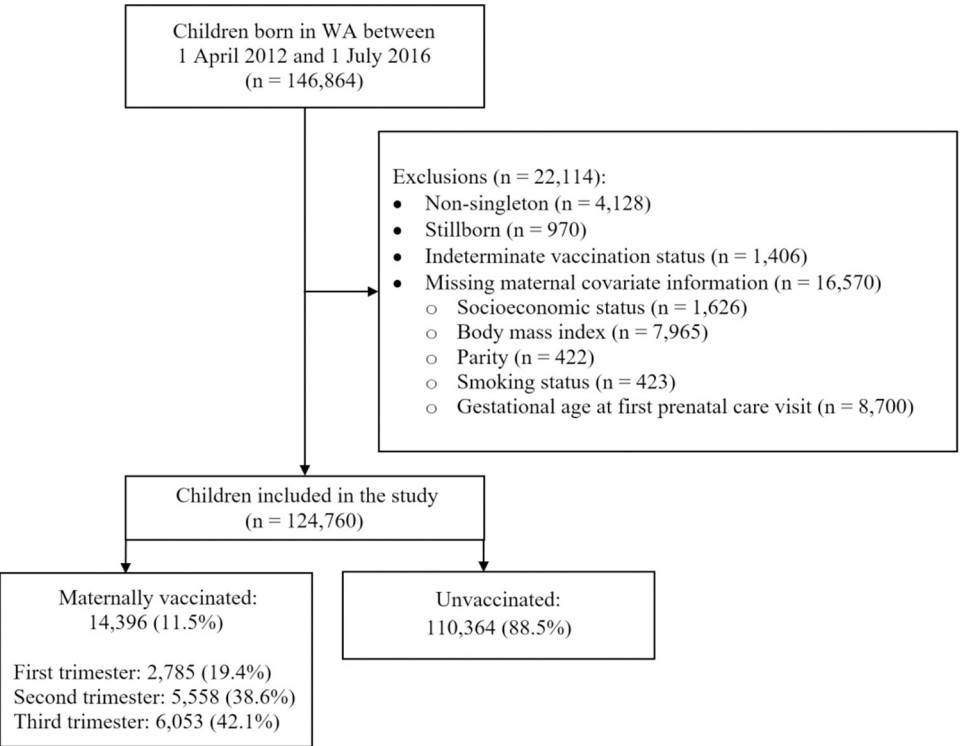

**Fig 1. Flow diagram of study participants included in the cohort.** WA, Western Australia.

compare common fields, such as given name, surname, date of birth, and other relevant fields (dependent on the contents and context of the dataset) [20] and provides a similarity weighting that is positively associated with the likelihood that 2 or more records belong to the same individual [19]. Clerical review is required to assess potential nonmatched records; this process has been shown to reduce the error rate of matching to less than 0.1% [21].

The MNS is a legally mandated perinatal data collection of all children born at least 20 weeks' gestation or birth weight of ≥400g (where gestational age is unknown). These data collection includes maternal demographic and health information, obstetric history, date of delivery, gestational age, and birth weight. Death registrations include the date and cause of all registered deaths in the state. This study received approval and a waiver or consent from the Department of Health WA Human Research Ethics Committee (RA#2016.56) and the Curtin University Human Research Ethics Committee (RA#20217–0808). This study is part of a larger linked cohort study and was guided by the study protocol described elsewhere [22]. This study is reported according to the REporting of studies Conducted using Observational Routinely-collected Data (RECORD) guideline (**S1 RECORD Checklist**) [23].

## Maternal influenza vaccination

Data on influenza vaccination during pregnancy were obtained from the WAAVD, a statewide database, managed by the WA Department of Health, which includes information on the date of vaccination, vaccine brand, and batch number, and the estimated gestation at which vaccinations were administered as reported by their healthcare provider. In Australia, seasonal IIV during pregnancy has been recommended by the Australian Government since March 2000 and funded under the National Immunisation Program since January 2010 [4,24].

Children of mothers who had a record of receipt of seasonal IIV during pregnancy were considered "maternally vaccinated." We estimated the gestational age at vaccination from the difference of completed weeks of gestation between the estimated date of conception and date of vaccination. Trimester at vaccination was categorized as first trimester (0 to $\leq$13 weeks gestation), second trimester (14 to $\leq$27 weeks gestation), and third trimester ($\geq$28 weeks gestation). As antibody transfer may have been suboptimal when administered <2 weeks prior to birth [25], we excluded children of mothers that received an influenza vaccine <2 weeks prior to birth (i.e., "indeterminate" vaccination status).

## Outcomes

Outcomes were identified from hospital and emergency department (ED) records [17,18]. The HMDC records all hospital inpatient admissions in the state's public and private hospitals, including the date of admission and separation, up to 21 diagnosis codes (classified according to International Statistical Classification of Diseases and Related Health Problems, Tenth Revision, Australian Modification [ICD-10-AM] codes) [17]. The EDDC includes all ED presentations in the state's metropolitan public and private hospitals, including the date of presentation, 1 ICD-10-AM diagnosis code, and 1 symptom code. Outcomes evaluated in this study included (a) allergic or autoimmune diseases; (b) allergic diseases; and (c) autoimmune diseases, as identified by ICD-10-AM diagnostic codes in primary or additional diagnosis fields (**S1 Table**). Within allergic diseases, we separately evaluated asthma or anaphylaxis as health outcomes. To evaluate the possible influence of bias in the study results, we included presentations for all-cause injuries as a negative control outcome, since negative controls can be used to identify noncausal associations between prenatal exposure to seasonal IIV and the outcomes (**S1 Table**) [26].

## Covariates

Maternal characteristics included age at the time of her child's birth ($\leq$19, 20 to 24, 25 to 29, 30 to 34, and $\geq$35 years), Aboriginal and/or Torres Strait Islander status (hereafter respectfully referred to as Aboriginal), socioeconomic status (quintiles between 1 (most disadvantaged) and 5 (least disadvantaged)) [27], body mass index, parity (primiparous, 1 prior birth, and $\geq$2 prior births), preexisting medical conditions (asthma, essential hypertension, and preexisting diabetes), pregnancy complications (gestational diabetes, gestational hypertension, and preeclampsia), smoking during pregnancy, and gestational age at the first prenatal care visit. Socioeconomic status was based on the Socioeconomic Index for Areas Index of Relative Socioeconomic Advantage and Disadvantage, an area-based index of relative access to resources for households within the same census collection district [27]. The referent group for socioeconomic status was the least disadvantaged quintile (i.e., quintile 5).

Child characteristics considered as covariates included the child's Aboriginal status, year and season of birth, preterm birth, and small-for-gestational age birth status. All covariates were selected *a priori* for their association with maternal influenza vaccination [28–33]. Preterm birth was categorized as moderate to late preterm (32 to <37 weeks gestation), very preterm (28 to <32 weeks gestation), and extremely preterm (<28 weeks gestation) birth. Small-for-gestational age birth was defined as birth weight <10th percentile according to Australian national birth weight percentiles by sex and gestational age [34].

## Participant involvement

No participants were involved in setting the research question or the exposure or outcome measures nor were they involved in the design and implementation of the study. There are no plans to directly involve study participants in the dissemination of the research findings.

## Statistical analyses

We compared the demographic and health characteristics between vaccinated and unvaccinated mothers and their children using univariate logistic regression models. Based on the predicted probability of vaccination from multivariate logistic regression, we estimated the inverse probability of treatment (vaccination) weights (IPTWs) to control for baseline probability of vaccination in further analyses. The multivariate model included the maternal covariates: age, Aboriginal status, socioeconomic status, body mass index, parity, preexisting medical conditions, pregnancy complications, smoking during pregnancy, and gestational age at the time of first prenatal care visit. To assess the balance of maternal covariates between maternally vaccinated and unvaccinated children, we calculated the standardized mean differences of each covariate. IPTWs were applied to Cox proportional hazards regression models to estimate unadjusted and adjusted hazard ratios (HRs) with 95% confidence intervals (CIs) for each of the study outcomes. Adjusted models additionally controlled for the child's Aboriginal status and year and season of birth.

Children were followed up from the date of birth and were censored at the earliest of (a) the date the child reached 5 years of age; (b) the last date of available data provided by the WA Data Linkage Branch (i.e., July 1, 2017); (c) the date the child died; or (d) the date of the event. Subgroup analyses compared the risk of study outcomes by trimester of maternal vaccination. To assess the sensitivity of our definition of childhood asthma, we restricted the definition of asthma to the presence of a diagnosis code for asthma alone (i.e., J45 and J46) and did not include early symptoms of asthma (i.e., wheezing). To evaluate associations with more severe clinical outcomes, additional analyses defined outcomes based on hospital inpatient admission data only. To evaluate the importance of timing of maternal vaccination and duration of exposure to maternal antibodies, stratified analyses were planned *a priori* and performed (1) by trimester of vaccination and additional sensitivity analysis stratifying (2) by preterm birth status, to see if results differed by length of gestation. All analyses were performed in Stata version 15.1 (Stata, College Station, Texas, United States of America).

As per recommendations during the peer review process, additional analyses were conducted. To evaluate the potential competing risk of all-cause mortality, the mortality rate of maternally vaccinated and unvaccinated children was calculated. As the majority of studies in maternal influenza vaccination and early childhood health outcomes have focused on the first 6 months of life, we performed a sensitivity analysis restricting to children aged between 6 months and 5 years. As the study cohort includes siblings, this may imply statistical dependence of observations due to genetic factors and/or shared environment; therefore, we performed an additional sensitivity analysis restricting to 1 randomly selected child per mother. To further evaluate the potential influence of seasonality on our findings, we performed an additional sensitivity analysis that matched children by calendar year and month of birth, allowing the assessment of outcomes comparing children born during similar time periods. These models incorporated year and month of birth as stratum and removed year and season of birth from propensity score estimates.

## Results

We identified a total of 146,864 children born in WA during the study period; 22,114 (15.1%) children were excluded because the child was a nonsingleton (*n* = 4,128), stillborn (*n* = 970), had indeterminate maternal vaccination status (*n* = 1,406), or missing covariate information (*n* = 16,570). The final cohort included 124,760 singleton, live-born children from 106,206 mothers (**Fig 1**). Children were followed until a mean age of 3.0 (standard deviation: 1.3)

years. The mortality rate was 2.4 deaths per 1,000 maternally vaccinated children and 2.4 deaths per 1,000 maternally unvaccinated children.

## Maternal influenza vaccination

Among the 124,760 children, 14,396 (11.5%) were maternally vaccinated: 2,785 (19.4%) exposed during the first trimester, 5,558 (38.6%) during the second trimester, and 6,053 (42.1%) during the third trimester. Vaccine exposure varied from 6.1% for births in 2012 to 12.0% in 2015. Most mothers received their influenza vaccine between March and July (*n* = 12,537; 87.1%) of any given year. Primiparous women, women with preexisting medical conditions and pregnancy complications, and women who were nonsmokers during pregnancy were more likely to have been vaccinated (**Table 1**). Vaccination was less common among women of the lowest socioeconomic status compared to women of the highest socioeconomic status (odds ratio [OR], 0.93; 95% CI, 0.88 to 0.98) and mothers of preterm infants compared to term infants (OR, 0.91; 95% CI, 0.85 to 0.98), and vaccination was more common among women who birthed during winter compared to summer (OR, 2.60; 95% CI, 2.46 to 2.74). After applying IPTW, standardized differences of maternal characteristics were balanced between maternally vaccinated and unvaccinated children (**S1 Fig**).

## Allergic or autoimmune diseases

During the study period, we identified 8,568 (6.9%) children with a diagnosis of an allergic and/or autoimmune disease. There was no significant difference in the risk of allergic or autoimmune diseases among maternally vaccinated children compared to maternally unvaccinated children (aHR, 1.02; 95% CI, 0.95 to 1.09) (**Table 2**). We identified no associations after stratifying by preterm birth status (**S2 and S3 Tables**) or when restricting to hospital inpatient admissions (**S4 Table**), children aged between 6 months and 5 years (**S5 Table**), or 1 random child per mother (**S6 Table**).

## Allergic diseases

There were 8,417 children with a diagnosis of an allergic disease. The most common outcomes included asthma (*n* = 3,538; 42.0%), urticaria (*n* = 2,805; 33.3%), and anaphylaxis (*n* = 987; 11.7%). Maternal influenza vaccination was not associated with presentation to ED or hospitalization for an allergic disease among children aged <5 years (aHR, 1.02; 95% CI, 0.95 to 1.10) (**Table 2**). We observed similar results when stratifying by preterm birth status (**S2 and S3 Tables**) and when restricting to hospital admissions only (**S4 Table**), children aged between 6 months and 5 years (**S5 Table**), or 1 random child per mother (**S6 Table**).

## Asthma

A total of 3,772 children had an episode of care for asthma: 1,020 (27.1%) diagnoses of asthma and 2,752 (73.3%) diagnoses of wheezing were identified. We observed no difference in the risk of asthma among maternally vaccinated children compared to maternally unvaccinated children (aHR, 1.00; 95% CI, 0.89 to 1.12) (**Table 2**).

In the sensitivity analysis restricting the definition of asthma to the presence of an asthma diagnosis code alone, we observed a negative association of asthma following vaccination during the third trimester (aHR, 0.70; 95% CI, 0.50 to 0.97) (**Table 2**). This was mostly attributed to a negative association observed among term children following vaccination during the third

**Table 1. Odds of seasonal influenza vaccination by maternal and child characteristics for children born in WA between April 1, 2012 and July 1, 2016.**

| Characteristic | Maternally unvaccinated (n = 110,364) n (%) | Maternally vaccinated (n = 14,396) n (%) | OR (95% CI) |
|---|---|---|---|
| *Maternal characteristics* | | | |
| Age (years) | | | |
| ≤19 | 3,369 (3.1) | 449 (3.1) | **1.13 (1.01 to 1.26)** |
| 20 to 24 | 14,828 (13.4) | 1,747 (12.1) | Ref |
| 25 to 29 | 31,461 (28.5) | 3,978 (27.6) | **1.07 (1.01 to 1.14)** |
| 30 to 34 | 37,706 (34.2) | 5,151 (35.8) | **1.16 (1.09 to 1.23)** |
| ≥35 | 23,000 (20.8) | 3,071 (21.3) | **1.13 (1.06 to 1.21)** |
| Aboriginal status | | | |
| Aboriginal | 5,296 (4.8) | 717 (5.0) | 1.04 (0.96 to 1.13) |
| Non-Aboriginal | 105,068 (95.2) | 13,679 (95.0) | Ref |
| Socioeconomic status[a] | | | |
| Quintile 1 (most disadvantaged) | 21,179 (19.2) | 2,618 (18.2) | **0.93 (0.88 to 0.98)** |
| Quintile 2 | 22,674 (20.5) | 3,019 (21.0) | 1.00 (0.94 to 1.05) |
| Quintile 3 | 23,257 (21.1) | 2,972 (20.6) | 0.96 (0.91 to 1.01) |
| Quintile 4 | 22,218 (20.1) | 2,979 (20.7) | 1.00 (0.95 to 1.06) |
| Quintile 5 (least disadvantaged) | 21,036 (19.1) | 2,808 (19.5) | Ref |
| Body mass index | | | |
| <18.5 (underweight) | 3,533 (3.2) | 450 (3.1) | 0.97 (0.87 to 1.07) |
| 18.5 to <25 (normal) | 54,032 (49.0) | 7,114 (49.4) | Ref |
| 25 to <30 (overweight) | 30,718 (27.8) | 3,875 (26.9) | 0.96 (0.92 to 1.00) |
| ≥30 (obese) | 22,081 (20.0) | 2,957 (20.5) | 1.02 (0.97 to 1.06) |
| Parity | | | |
| Primiparous | 47,823 (43.3) | 6,764 (47.0) | Ref |
| 1 prior birth | 38,216 (34.6) | 4,958 (34.4) | **0.92 (0.88 to 0.95)** |
| ≥2 prior births | 24,325 (22.0) | 2,674 (18.6) | **0.78 (0.74 to 0.81)** |
| Preexisting medical conditions | | | |
| Asthma | 11,522 (10.4) | 1,596 (11.1) | **1.07 (1.01 to 1.13)** |
| Essential hypertension | 1,417 (1.3) | 261 (1.8) | **1.42 (1.24 to 1.62)** |
| Preexisting diabetes mellitus | 916 (0.8) | 191 (1.3) | **1.61 (1.37 to 1.88)** |
| Pregnancy complications | | | |
| Gestational diabetes | 11,310 (10.3) | 1,688 (11.7) | **1.16 (1.10 to 1.23)** |
| Gestational hypertension | 5,111 (4.6) | 777 (5.4) | **1.17 (1.09 to 1.27)** |
| Preeclampsia | 3,596 (3.3) | 543 (3.8) | **1.16 (1.06 to 1.28)** |
| Smoked during pregnancy | 10,541 (9.6) | 1,269 (8.8) | **0.92 (0.86 to 0.97)** |
| Trimester of first prenatal care visit | | | |
| First trimester | 72,861 (66.0) | * | Ref |
| Second trimester | 32,384 (29.3) | * | **0.82 (0.79 to 0.85)** |
| Third trimester | 5,053 (4.6) | * | **0.54 (0.49 to 0.60)** |
| No prenatal care | 66 (0.1) | <5 | 0.43 (0.16 to 1.18) |
| Year of birth | | | |
| 2012 | 19,922 (18.1) | 1,304 (9.1) | Ref |
| 2013 | 25,898 (23.5) | 2,909 (20.2) | **1.72 (1.60 to 1.84)** |
| 2014 | 26,347 (23.9) | 3,205 (22.3) | **1.86 (1.74 to 1.99)** |
| 2015 | 24,727 (22.4) | 5,148 (35.8) | **3.18 (2.98 to 3.39)** |
| 2016 | 13,470 (12.2) | 1,830 (12.7) | **2.08 (1.93 to 2.24)** |
| Season of birth | | | |

(*Continued*)

**Table 1.** (Continued)

| Characteristic | Maternally unvaccinated (n = 110,364) n (%) | Maternally vaccinated (n = 14,396) n (%) | OR (95% CI) |
|---|---|---|---|
| Summer (December to February) | 27,327 (24.8) | 2,186 (15.2) | Ref |
| Autumn (March to May) | 31,718 (28.7) | 2,291 (15.9) | **0.90 (0.85 to 0.96)** |
| Winter (June to August) | 26,056 (23.6) | 5,413 (37.6) | **2.60 (2.46 to 2.74)** |
| Spring (September to November) | 25,263 (22.9) | 4,506 (31.3) | **2.23 (2.11 to 2.35)** |
| *Child characteristics* | | | |
| Sex[b] | | | |
| Male | 56,820 (51.5) | 7,347 (51.0) | Ref |
| Female | 53,541 (48.5) | 7,049 (49.0) | 1.02 (0.98 to 1.05) |
| Aboriginal status | | | |
| Aboriginal | 5,779 (5.2) | 787 (5.5) | 1.05 (0.97 to 1.13) |
| Non-Aboriginal | 104,585 (94.8) | 13,609 (94.5) | Ref |
| Birth outcomes | | | |
| Preterm birth | 7,390 (6.7) | 887 (6.1) | **0.91 (0.85 to 0.98)** |
| Moderate-to-late preterm | 6,587 (6.0) | 819 (5.7) | 0.95 (0.88 to 1.02) |
| Very preterm | 520 (0.5) | 52 (0.4) | 0.76 (0.57 to 1.01) |
| Extremely preterm | 291 (0.3) | 16 (0.1) | **0.42 (0.25 to 0.69)** |
| Small-for-gestational age[c] | 9,006 (8.2) | 1,191 (8.3) | 1.02 (0.95 to 1.08) |

[a] Socioeconomic status was based on the Socioeconomic Index for Areas measure of relative socioeconomic advantage and disadvantage developed by the Australian Bureau of Statistics [27].

[b] The sex of <5 maternally unvaccinated children was unknown.

[c] Small-for-gestational age was based on the Australian national birth weight percentiles by sex and gestational age [34].

* In accordance with privacy and confidentiality guidelines by the WA Data Linkage Branch, secondary suppression was used to prevent suppressed cells (<5) from being recalculated through subtraction.

CI, confidence interval; OR, odds ratio; WA, Western Australia.

trimester (aHR, 0.68; 95% CI, 0.48 to 0.96) (S2 Table). When considering year and month of birth as a stratum, this negative association between asthma and vaccination during the third trimester was no longer significant (S7 Table). No other associations were observed following vaccination during earlier trimesters (Table 2) or after stratifying by preterm birth status (S2 and S3 Tables) or when restricting to hospital admissions only (S4 Table), children aged between 6 months and 5 years (S5 Table), or 1 random child per mother (S6 Table).

## Anaphylaxis

In total, there were 1,157 children diagnosed with an episode of anaphylaxis. We observed a negative association of anaphylaxis associated with exposure to seasonal IIV administered during the third trimester (aHR, 0.67; 95% CI, 0.47 to 0.95). This was attributed to a negative association observed among term children following vaccination during the third trimester (aHR, 0.62; 95% CI, 0.43 to 0.91) (S2 Table). The negative association between anaphylaxis and vaccination during the third trimester remained even after treating year and month of birth as a stratum (aHR, 0.69; 95% CI, 0.49 to 0.97) (S7 Table). No other associations were observed following vaccination during earlier trimesters (Table 2) or after stratifying by preterm birth status (S2 and S3 Tables) or when restricting to hospital admissions only (S4 Table), children aged between 6 months and 5 years (S5 Table), or 1 random child per mother (S6 Table).

**Table 2. Risk of allergic or autoimmune diseases associated with prenatal exposure to seasonal inactivated influenza vaccine among children <5 years of age, by trimester of prenatal vaccination.**

| | Unexposed to seasonal influenza vaccine during pregnancy (N = 110,364) | Exposed to seasonal influenza vaccine during pregnancy (N = 14,396) | Trimester of vaccine exposure | | |
| --- | --- | --- | --- | --- | --- |
| | | | First trimester (N = 2,785) | Second trimester (N = 5,558) | Third trimester (N = 6,053) |
| *Allergic or autoimmune disease* | | | | | |
| Cases, n (%) | 7,655 (6.9) | 913 (6.3) | 171 (6.1) | 395 (7.1) | 347 (5.7) |
| Unweighted HR (95% CI) | 1 [Reference] | 1.03 (0.96 to 1.11) | 0.99 (0.85 to 1.15) | 1.09 (0.99 to 1.21) | 0.99 (0.89 to 1.11) |
| Weighted aHR (95% CI)[a] | 1 [Reference] | 1.02 (0.95 to 1.09) | 0.97 (0.82 to 1.14) | 1.07 (0.96 to 1.19) | 0.98 (0.88 to 1.10) |
| *Allergic disease* | | | | | |
| Cases, n (%) | 7,518 (6.8) | 899 (6.2) | 168 (6.0) | 391 (7.0) | 340 (5.6) |
| Unweighted HR (95% CI) | 1 [Reference] | 1.03 (0.96 to 1.11) | 0.99 (0.85 to 1.15) | 1.10 (0.99 to 1.22) | 0.99 (0.89 to 1.10) |
| Weighted aHR (95% CI)[a] | 1 [Reference] | 1.02 (0.95 to 1.10) | 0.97 (0.82 to 1.14) | 1.08 (0.97 to 1.20) | 0.98 (0.87 to 1.10) |
| *Asthma diagnosis or wheezing* | | | | | |
| Cases, n (%) | 3,375 (3.1) | 382 (2.7) | 68 (2.4) | 169 (3.0) | 145 (2.4) |
| Unweighted HR (95% CI) | 1 [Reference] | 1.01 (0.91 to 1.12) | 0.92 (0.72 to 1.17) | 1.07 (0.91 to 1.25) | 1.00 (0.84 to 1.18) |
| Weighted aHR (95% CI)[a] | 1 [Reference] | 1.00 (0.89 to 1.12) | 0.92 (0.71 to 1.19) | 1.07 (0.91 to 1.26) | 0.97 (0.81 to 1.15) |
| *Asthma diagnosis only*[b] | | | | | |
| Cases, n (%) | 1,425 (1.3) | 131 (0.9) | 30 (1.1) | 61 (1.1) | 40 (0.7) |
| Unweighted HR (95% CI) | 1 [Reference] | 0.89 (0.74 to 1.06) | 1.03 (0.72 to 1.49) | 0.97 (0.75 to 1.25) | **0.72 (0.52 to 0.98)** |
| Weighted aHR (95% CI)[a] | 1 [Reference] | 0.87 (0.73 to 1.05) | 0.99 (0.68 to 1.45) | 0.98 (0.75 to 1.28) | **0.70 (0.50 to 0.97)** |
| *Anaphylaxis* | | | | | |
| Cases, n (%) | 1,043 (0.9) | 114 (0.8) | 30 (1.1) | 48 (0.9) | 36 (0.6) |
| Unweighted HR (95% CI) | 1 [Reference] | 0.95 (0.78 to 1.15) | 1.29 (0.90 to 1.85) | 0.98 (0.74 to 1.31) | 0.75 (0.54 to 1.05) |
| Weighted aHR (95% CI)[a] | 1 [Reference] | 0.85 (0.70 to 1.05) | 1.15 (0.78 to 1.68) | 0.90 (0.67 to 1.22) | **0.67 (0.47 to 0.95)** |
| *Autoimmune disease* | | | | | |
| Cases, n (%) | 158 (0.1) | 16 (0.1) | <5 | 6 (0.1) | 7 (0.1) |
| Unweighted HR (95% CI) | 1 [Reference] | 0.95 (0.57 to 1.59) | - | 0.85 (0.37 to 1.92) | 1.08 (0.50 to 2.30) |
| Weighted aHR (95% CI)[a] | 1 [Reference] | 0.93 (0.55 to 1.59) | - | 0.89 (0.37 to 2.12) | 1.06 (0.49 to 2.31) |

All outcomes were identified from ICD-10-AM codes found in the principal and additional diagnosis fields of hospital inpatient and ED presentation records and from the presenting symptom code found in the ED presentation records (**S1 Table**).

[a] HRs were weighted by inverse probability of treatment factoring for maternal covariates including age, Aboriginal status, socioeconomic status, body mass index, parity, preexisting medical conditions (asthma, essential hypertension, and preexisting diabetes), pregnancy complications (gestational diabetes, gestational hypertension, and preeclampsia), smoking status during pregnancy, gestational age at first prenatal care visit, year and season of birth; models were additionally adjusted for child's Aboriginal status.

[b] Sensitivity analysis restricting the definition of asthma to the presence of a diagnosis code of asthma alone (i.e., J45 and J46).

-, indeterminate (a stable estimate could not be generated due to the low number of outcomes); aHR, adjusted hazard ratio; CI, confidence interval; ED, emergency department; HR, unadjusted hazard ratio; ICD-10-AM, International Statistical Classification of Diseases and Related Health Problems, Tenth Revision, Australian Modification.

## Autoimmune diseases

There were 174 children with a diagnosis of an autoimmune disease, including 16 (9.2%) who were maternally vaccinated. The most common outcomes included diabetes mellitus (*n* = 54; 31.0%), coeliac disease (*n* = 40; 23.0%), idiopathic thrombocytopenic purpura (*n* = 33; 19.0%), and juvenile arthritis (*n* = 21; 12.1%). There was no significant difference in the risk of autoimmune diseases between maternally vaccinated and unvaccinated children (aHR, 0.93; 95% CI, 0.55 to 1.59) (**Table 2**) nor any differences detected by trimester of vaccination (**Table 2**), after stratifying by preterm birth status (**S2 and S3 Tables**), or when restricting to hospital

admissions only (S4 Table), children aged between 6 months and 5 years (S5 Table), or 1 random child per mother (S6 Table).

## Negative control

A total of 21,730 children aged <5 years presented to ED or were hospitalized with an injury. We observed no difference in the risk of all-cause injuries between maternally vaccinated and unvaccinated children (aHR, 1.04; 95% CI, 0.99 to 1.11) (S8 Table).

## Discussion

In this large, population-based cohort study of 124,760 children, we found no evidence of an increased risk of allergic or autoimmune disease in children following prenatal exposure to seasonal IIV. We observed a negative association of anaphylaxis and some indications of lower risk of asthma following seasonal IIV given during the third trimester. We observed no other associations between prenatal exposure to seasonal IIV and allergic or autoimmune diseases in children up to 5 years of age. These results contribute to the gap in knowledge of the potential child health impacts of maternal influenza vaccination on the development of allergic or autoimmune diseases in childhood and support the safety and continuation of existing maternal vaccination programs and policies.

To date, most studies evaluating maternal influenza vaccination have focused on outcomes in the first 6 months of life. To our knowledge, only 3 studies have assessed allergic and/or autoimmune outcomes in children beyond 6 months of age [11–13]. First, a recent Canadian study by Mehrabadi and colleagues [11] examined asthma in children aged <6 years, born between October 2010 and March 2014, following prenatal exposure to unadjuvanted trivalent (i.e., 2 type A strains and 1 type B strain) seasonal influenza vaccine. Another Canadian study by Walsh and colleagues [12] examined asthma in children aged <5 years, born between November 2009 and October 2010, following prenatal exposure to nonadjuvanted or AS03-adjuvanted pandemic A/H1N1 influenza vaccine. A Danish study by Hviid and colleagues [13] also examined asthma as well as other autoimmune diseases in children aged <5 years, born between November 2009 and March 2010, following prenatal exposure to AS03-adjuvanted pandemic A/H1N1 influenza vaccine. In this study, stratified analyses of outcomes were restricted to (a) first trimester exposure; and (b) second- or third-trimester exposure to vaccine. In Denmark, only pregnant women with chronic diseases were recommended to get vaccinated during this period, resulting in a low number of children vaccinated during the first trimester (n = 349; 5.5%) and therefore likely lacked statistical power to detect associations. These studies identified no association between maternal influenza vaccination and the subsequent development of allergic or autoimmune diseases in offspring.

While our results evaluating seasonal influenza vaccine mostly align with those from prior pandemic vaccine studies, when stratifying by trimester of vaccination, we observed a negative association of asthma and anaphylaxis following third trimester influenza vaccination. This association persisted in a sensitivity analysis restricting to term children but not in preterm children.

While we cannot entirely rule out the possible influence of bias in the results from this observational study, several aspects of our study suggest a plausible protective relationship between maternal immunization and the development of allergic conditions in childhood. First, early exposure to respiratory infections, in particular viral infections, are strongly linked to the development of childhood wheezing disorders and subsequent asthma in susceptible children [35,36], and a history of asthma is a known risk factor of anaphylaxis and severe anaphylactic episodes, not surprisingly, as there is interaction between atopic asthma

and other allergic states [37]. Maternal influenza immunization reduces the risk of all-cause respiratory infection in infants <6 months by 25% [38,39]. Protection from respiratory infection through maternal immunization could help to avoid exacerbation of wheezing episodes through recurrent infections and subsequent development of asthma. Given maternal antibody transfer peaks during the third trimester of pregnancy [40], children born at term are likely to receive more maternal antibodies, and thus a higher level of protection, than children born preterm. This is consistent with our observation that a negative association of asthma was only observed among maternally vaccinated children born term. Second, priming of the infant's developing immune system may occur in response to exposure to environmental allergens, maternal vaccination, or infection [9]. Studies have shown an adaptive antigen-specific cellular immune response independent of antibody-mediated passive immunity associated with maternal vaccination, which suggests potential nonspecific effects of maternal influenza vaccination [9].

Our study had several strengths and limitations. The main strengths of our study include the use of a large, population-based mother–infant cohort with detailed information on maternal sociodemographic and health characteristics, receipt of prenatal vaccine, and record linkage to several administrative health datasets permitting follow-up of outcomes up to 5 years of age. The record linkage system in WA is long-standing, with expertise in linking large administrative health data since 1995 [41]. With the exception of the antenatal vaccination register, these datasets are legally mandated, feed into national data collections, and the quality of data is considered to be high [42]. The MNS is also estimated to capture 99% of all births in WA [15].

Despite these strengths, our study has several limitations. First, maternal vaccination data linked from the WAAVD relied on immunization reports from medical providers and, although these records have high specificity, they are likely to be incomplete [16]. Second, outcomes were limited to diagnoses recorded in hospital inpatient and ED presentation records and do not capture outcomes diagnosed and treated in a primary care setting. However, because severe medical events, such as anaphylaxis, are more likely to present for secondary or tertiary care, the more severe outcomes we assessed in this study were likely to be well measured. For this reason, we considered different definitions of study outcomes to evaluate the sensitivity of diagnostic codes. Third, we cannot entirely rule out residual confounding, although we attempted to restrict the influence of health-seeking behavior using inverse probability of treatment weighting. Finally, despite the large size of our cohort, small cell sizes (i.e., <5) made it impractical to estimate effects for first trimester or by preterm birth status for all outcomes. To our knowledge, only one other study has evaluated the effect of seasonal IIV during pregnancy on allergic or autoimmune disease development in early childhood. Overall, our findings suggest that seasonal IIV during pregnancy is not associated with adverse allergic or autoimmune outcomes in children aged up to 5 years and support current global vaccine policies prioritizing influenza immunization for pregnant women. Our study observed a negative association of anaphylaxis and asthma following seasonal IIV given during the third trimester, and these results warrant further investigation. This information is useful to pregnant women and their healthcare providers when making vaccine decisions and providing vaccine counseling.

## Supporting information

**S1 RECORD Checklist. The RECORD statement—checklist of items, extended from the STROBE statement, that should be reported in observational studies using routinely collected health data.** RECORD, REporting of studies Conducted using Observational

Routinely-collected Data.
(DOCX)

**S1 Table. ICD-10-AM diagnosis codes used to identify allergic or autoimmune diseases, and all-cause injuries, and frequency of outcomes by data source.** ICD-10-AM, International Statistical Classification of Diseases and Related Health Problems, Tenth Revision, Australian Modification.
(DOCX)

**S2 Table. Risk of allergic or autoimmune diseases associated with prenatal exposure to seasonal inactivated influenza vaccine among children <5 years of age who were born at term (>37 weeks gestational age), by trimester of prenatal vaccination.**
(DOCX)

**S3 Table. Risk of allergic or autoimmune diseases associated with prenatal exposure to seasonal inactivated influenza vaccine among children <5 years of age who were born preterm (<37 weeks gestational age), by trimester of prenatal vaccination.**
(DOCX)

**S4 Table. Risk of inpatient admission (only) for allergic or autoimmune disease associated with prenatal exposure to seasonal inactivated influenza vaccine among children <5 years of age, by trimester of prenatal vaccination.**
(DOCX)

**S5 Table. Risk of allergic or autoimmune diseases associated with prenatal exposure to seasonal inactivated influenza vaccine among children between 6 months and <5 years of age, by trimester of prenatal vaccination.**
(DOCX)

**S6 Table. Risk of allergic or autoimmune diseases associated with prenatal exposure to seasonal inactivated influenza vaccine among one randomly selected child per mother, by trimester of prenatal vaccination.**
(DOCX)

**S7 Table. Risk of allergic or autoimmune diseases associated with prenatal exposure to seasonal inactivated influenza vaccine among children <5 years of age matched by year and month of birth, by trimester of prenatal vaccination.**
(DOCX)

**S8 Table. Risk of all-cause injury associated with prenatal exposure to seasonal inactivated influenza vaccination among children <5 years of age, by trimester of prenatal vaccination.**
(DOCX)

**S1 Fig.** Balance of standardized differences of maternal covariates before and after inverse probability treatment weighting, by trimester of vaccination: **(a)** maternally vaccinated during any trimester; **(b)** maternally vaccinated during the first trimester; **(c)** maternally vaccinated during the second trimester; or **(d)** maternally vaccinated during the third trimester.
(TIF)

**S1 Appendix. Sample STATA code for running unweighted/unadjusted and weighted/adjusted models to estimate the association between prenatal exposure to seasonal inactivated influenza vaccine and allergic and/or autoimmune diseases in children.**
(DOCX)

## Acknowledgments

The authors would like to thank the Linkage and Client Services Team at the Data Linkage Branch, WA Department of Health, as well as the data custodians of the Birth Registrations, Midwives Notification System, WA Antenatal Vaccination Database, Hospital Morbidity Data Collection, Emergency Department Data Collection, and Death Registrations.

This study is based in part on data provided by the WA Data Linkage Branch, part of the WA Department of Health.

**Disclaimers**

The analyses, results, interpretation, and conclusions reported in this manuscript are those of the authors and do not necessarily represent those of the WA Data Linkage Branch.

## Author Contributions

**Conceptualization:** Damien Foo.

**Data curation:** Damien Foo, Mohinder Sarna, Annette K. Regan.

**Formal analysis:** Damien Foo.

**Funding acquisition:** Annette K. Regan.

**Methodology:** Damien Foo, Mohinder Sarna, Gavin Pereira, Hannah C. Moore, Annette K. Regan.

**Project administration:** Mohinder Sarna, Annette K. Regan.

**Supervision:** Mohinder Sarna, Gavin Pereira, Hannah C. Moore, Annette K. Regan.

**Validation:** Annette K. Regan.

**Visualization:** Damien Foo.

**Writing – original draft:** Damien Foo.

**Writing – review & editing:** Damien Foo, Mohinder Sarna, Gavin Pereira, Hannah C. Moore, Annette K. Regan.

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
