## [Editor Report · Decision Letter 0]

30 Jun 2021

Dear Dr Foo, 

Thank you for submitting your manuscript entitled "Prenatal influenza vaccination and allergic and autoimmune diseases in childhood: Longitudinal, population-based cohort study" for consideration by PLOS Medicine.

Your manuscript has now been evaluated by the PLOS Medicine editorial staff and I am writing to let you know that we would like to send your submission out for external peer review.

Kind regards,

Louise Gaynor-Brook, MBBS PhD

Senior Editor

PLOS Medicine

---

## [Decision Letter · Decision Letter 1]

18 Sep 2021

Dear Dr. Foo,

Thank you very much for submitting your manuscript "Prenatal influenza vaccination and allergic and autoimmune diseases in childhood: Longitudinal, population-based cohort study" (PMEDICINE-D-21-02861R1) for consideration at PLOS Medicine. 

Your paper was evaluated by three independent reviewers, including a statistical reviewer, and was discussed among all the editors here and with an academic editor with relevant expertise.The reviews are appended at the bottom of this email and any accompanying reviewer attachments can be seen via the link below:

[LINK]

In light of these reviews, I am afraid that we will not be able to accept the manuscript for publication in the journal in its current form, but we would like to consider a revised version that addresses the reviewers' and editors' comments. Obviously we cannot make any decision about publication until we have seen the revised manuscript and your response, and we plan to seek re-review by one or more of the reviewers. 

We expect to receive your revised manuscript by Oct 11 2021 11:59PM. Please email us (plosmedicine@plos.org) if you have any questions or concerns.

We look forward to receiving your revised manuscript. 

Sincerely,

Louise Gaynor-Brook, MBBS PhD

Associate Editor 

PLOS Medicine

plosmedicine.org

General comments:

Throughout the paper, please adapt reference call-outs to the following style: "... <5 years [1,2]" (noting the absence of spaces within the square brackets).

Please remove numeration of headings and subheadings 

Please speak of associations rather than “ higher/lower risk of” to avoid inadvertent causal language.

Data availability:

PLOS Medicine requires that the de-identified data underlying the specific results in a published article be made available, without restrictions on access, in a public repository or as Supporting Information at the time of article publication, provided it is legal and ethical to do so. 

If the data are owned by a third party but freely available upon request, please note this and state the owner of the data set and contact information for data requests (web or email address). Note that a study author cannot be the contact person for the data.

If the data are not freely available, please describe briefly the ethical, legal, or contractual restriction that prevents you from sharing it. Please also include an appropriate contact (web or email address) for inquiries (again, this cannot be a study author).

Title: Please revise your title according to PLOS Medicine's style. Please place the study design in the subtitle (ie, after a colon). We suggest “Prenatal influenza vaccination and allergic and autoimmune diseases in childhood: A population-based longitudinal cohort study” or similar

Abstract:

Abstract Methods and Findings:

Please provide brief demographic details of the study population (e.g. sex, age, ethnicity, etc)

Please define the length of follow up (eg, in mean, SD, and range).

Please include the important dependent variables that are adjusted for in the analyses.

For the trimester-specific analyses, please provide the actual numbers in addition to aHRs

In the last sentence of the Abstract Methods and Findings section, please describe 2-3 of the main limitations of the study's methodology.

Abstract Conclusions:

Please begin your Abstract Conclusions with "In this study, we observed ..." or similar, to summarize the main findings from your study, without overstating your conclusions. Please emphasize what is new and address the implications of your study, being careful to avoid assertions of primacy. 

Line 58 - Please speak of associations rather than “ lower risk of” to avoid inadvertent causal language.

Author Summary:

In the final bullet point of ‘What Do These Findings Mean?’, please describe the main limitations of the study in non-technical language.

Introduction:

Please expand upon the potential importance of your study. Please indicate whether your study is novel and how you determined that, being careful to temper assertions of primacy. If there has been a systematic review of the evidence related to your study (or you have conducted one), please refer to and reference that review and indicate whether it supports the need for your study. 

Methods:

Did your study have a prospective protocol or analysis plan? Please state this (either way) early in the Methods section. If a prospective analysis plan (from your funding proposal, IRB or other ethics committee submission, study protocol, or other planning document written before analyzing the data) was used in designing the study, please include the relevant prospectively written document with your revised manuscript as a Supporting Information file to be published alongside your study, and cite it in the Methods section. A legend for this file should be included at the end of your manuscript. If no such document exists, please make sure that the Methods section transparently describes when analyses were planned, and if/when reported analyses differed from those that were planned. Changes in the analysis-- including those made in response to peer review comments-- should be identified as such in the Methods section of the paper, with rationale. If a reported analysis was performed based on an interesting but unanticipated pattern in the data, please be clear that the analysis was data-driven.

Please ensure that the study is reported according to the RECORD guideline, and include the completed RECORD checklist as Supporting Information. Please add the following statement, or similar, to the Methods: "This study is reported as per the REporting of studies Conducted using Observational Routinely-collected Data (RECORD) guideline (S1 Checklist)." The RECORD guideline can be found here: https://www.record-statement.org/checklist.php When completing the checklist, please use section and paragraph numbers, rather than page numbers which will likely no longer correspond to the appropriate sections after copy-editing.

Results: 

For the OR relating to lowest socioeconomic status, please specify the comparison group in the main text.

Please define the length of follow up (eg, in mean, SD, and range).

Discussion:

Please present and organize the Discussion as follows: a short, clear summary of the article's findings; what the study adds to existing research and where and why the results may differ from previous research; strengths and limitations of the study; implications and next steps for research, clinical practice, and/or public policy; one-paragraph conclusion.

Please remove all subheadings within your Discussion i.e. Conclusion

References:

Please ensure that journal name abbreviations match those found in the National Center for Biotechnology Information (NCBI) databases, and are appropriately formatted and capitalised.

Please also see https://journals.plos.org/plosmedicine/s/submission-guidelines#loc-references for further details on reference formatting. 

Where websites are cited, please specify the date of access

Comments from the reviewers:

Reviewer #1: This is a well-conducted population-based study on the association between prenatal influenza vaccination and allergic and autoimmune diseases in childhood. The study design, datasets, statistical methods and analyses, and presentation (tables and figues) and interpretation of results are mostly adequate. However, there are still a few issues needing attention.

1) Table 1 on vaccination comparison. Odds ratios from univariate logistic regression are not informative and could be misleading as not multiple-adjusted. Suggest to replace ORs with simple P-values from Chi-squared test as the percentages themselves in the table will give the main information for comparison.

2) The post-hoc power analysis (page 9) is inadequate and redundant for this type of population-based observational studies. Suggest to remove it from the paper including the S2 Table.

3) What's the mortality rate in this cohort? As the outcome of the Cox models are allergy and autoimmune diseases other than all-cause mortality, there is a potential competing risk from death in the survival analyses. I suspect the mortality rate is low but need to mention it so that we can make sure there is little chance of competing risk in the analyses.

Reviewer #2: Dr. Foo and co-authors have evaluated a topic most interesting for both science community and general public. The main claim in the manuscript is that influenza vaccination given to pregnant mothers is not associated with risk of allergic or autoimmune diseases in the offspring by five years of age. This result will help public health authorities in their effort to promote influenza vaccination for pregnant women.

Influenza vaccination coverage among pregnant women was only 6.1 - 12.0% during the calendar years included in the study. Still, thanks to the large population-based cohort setting the statistical power was adequate. Statistical analyses included also a comprehensive covariate analysis.

The manuscript is easy to read even for non-specialists.

My only criticism concerns identification of allergic and autoimmune diseases in the study population. Data on these condition was retrieved from emergency department (ED) records and hospital inpatient admission records. This is a big problem especially for allergic diseases and perhaps to minor extent also for autoimmune diseases. Let's think of the allergic diseases first. Surely children with common allergic diseases like hay fever or allergy to foods or animals hardly ever need emergency care or hospitalization due to these conditions. Also, children with diagnosed asthma don't need emergency care or hospitalization unless their asthma is seriously off-balance. It is therefore evident that this study captures only a minority of the allergic diseases in the population. The authors can certainly claim that influenza vaccination is not associated with these severe cases of allergic disease, but it's perhaps too straightforward to claim no association with the vast bulk of allergic diseases. The authors try to discuss this limitation, and they quite correctly state that "..outcomes were limited by diagnoses recorded in hospital inpatient and ED presentation

records, which do not capture outcomes diagnosed and treated in a primary care setting." (lines 321-2) However, in the next sentences (lines 323-6) I don't quite follow their line of thinking. I suggest that the authors re-phrase this part of the limitations section and perhaps it would be wise to emphasize in the abstract/conclusions that the study concentrates on the most severe allergic conditions requiring ED or hospitalization, and the results can not necessarily be generalized for all allergic diseases.

The same criticism concerns autoimmune diseases, though to much lesser extent. The most common autoimmune diseases (diabetes, coeliac disease, idiopathic thrombocytopenic purpura and juvenile arthritis) quite often require ED care or hospitalization especially at the time of diagnosis. Still, the authors could discuss this a bit. Perhaps some of these conditions (coeliac disease?) can be diagnosed and managed in out-patient setting, and these cases are not captured in this study? Perhaps I'm wrong here, as I'm not fully aware how Australian health care system works.

Reviewer #3: I find the manuscript to be well written and generally well performed. The size of the studied cohort yield valuable results contributing to the quantification of the relative safety of influenza vaccination during pregnancy with respect to more long-term outcomes in the children.

As stated by the authors most studies have focused on outcomes in the first six months of life. The results of this study are presented as the first occurrence of diagnoses up to five years of age, and it is therefore not clear to me how much events occurring during the first six months of life contributes to the estimated risks. I suggest that additional estimates as a function of attained age are presented, at least as supplemental tables (which also would address the proportional hazard assumption inherent in Cox' regression). Possibly the authors could also present analyses where admission/ER-visits during the first 6 months of life is disregarded, as a way of high-lighting the estimated risk beyond six months of age.

Minor

On row 174 it is written that subjects are censored at "the last date of available data". Could this be clarified? Is this a fixed date relating to how updated the hospital registers are, or is it relating to the migration from region?

Are there siblings included in the study cohort? This may imply statistical dependence between observations due to genetic factors and/or shared environment. One could consider a sensitivity analysis where each mother only contributes with one delivery, or otherwise control for this possible dependence.

[LINK]

---

## [Decision Letter · Decision Letter 2]

1 Dec 2021

Dear Dr. Foo,

Thank you very much for submitting your manuscript "Prenatal influenza vaccination and allergic and autoimmune diseases in childhood: A longitudinal, population-based linked cohort study" (PMEDICINE-D-21-02861R2) for consideration at PLOS Medicine. 

Your paper was evaluated by the statistical reviewer, and discussed among all the editors here and with an academic editor with relevant expertise. The reviews are appended at the bottom of this email and any accompanying reviewer attachments can be seen via the link below:

[LINK]

In light of the comments from the Academic Editor, I am afraid that we will not be able to accept the manuscript for publication in the journal in its current form, but we would like to consider a revised version that addresses the reviewers' and editors' comments. Obviously we cannot make any decision about publication until we have seen the revised manuscript and your response, and we plan to seek re-review by one or more of the reviewers. 

We expect to receive your revised manuscript by Dec 22 2021 11:59PM. Please email us (plosmedicine@plos.org) if you have any questions or concerns.

We look forward to receiving your revised manuscript. 

Sincerely,

Louise Gaynor-Brook, MBBS PhD

Associate Editor, PLOS Medicine

plosmedicine.org

Comments from the Academic Editor:

The authors found a protective association between immunization in third trimester and asthma and anaphylaxis. My concern was that there could be a risk of residual confounding. The reason is that in most settings there is an association between season at birth and allergic disorders. Influenza vaccination has also a seasonal variation. Third trimester vaccination implies that birth happens soon after vaccination - in the same season or the next. The authors could easily show whether there is a risk of residual confounding - by reporting whether asthma and anaphylaxis are associated with season of birth or not.

I observe another problem that easily could be fixed: the association between third trimester vaccination and asthma and anaphylaxis is presented as "a strong protective association" in author summary bullets. In Discussion the authors write: "We observed a negative association of anaphylaxis and some indications of low risk of asthma following seasonal IIV given during the third trimester." I believe the author summary bullet should be modified in line with the more careful extression they have in Discussion.

We suggest conducting a sensitivity analysis to assess the potential impact of season at birth on the associations you have observed in your study. This will be subject to statistical re-review. 

Comments from the reviewers:

Reviewer #1: Thanks authors for their effort to improve the manuscript. I am satisfied with the response and revision. No further issues needing attention.

[LINK]

---

## [Decision Letter · Decision Letter 3]

23 Feb 2022

Dear Dr. Foo,

Thank you very much for re-submitting your manuscript "Prenatal influenza vaccination and allergic and autoimmune diseases in childhood: A longitudinal, population-based linked cohort study" (PMEDICINE-D-21-02861R3) for review by PLOS Medicine.

I have discussed the paper with my colleagues and the academic editor and it was also seen again by one reviewer. I am pleased to say that provided the remaining editorial and production issues are dealt with we are planning to accept the paper for publication in the journal.

[LINK]

We look forward to receiving the revised manuscript by Mar 02 2022 11:59PM.   

Sincerely,

Louise Gaynor-Brook, MBBS PhD

PLOS Medicine

plosmedicine.org

Requests from Editors:

Data availability:

Please note that a study author cannot be a contact person for programming code. Please provide an alternative appropriate contact (web or email address) for inquiries, or consider publishing your code in GitHub (please provide a link). 

Abstract Methods and Findings:

Please define CI at first use

Abstract Conclusions:

Please address the implications of your study (as you have done in your Author Summary) .

Introduction:

Line 118 - Please reference the three studies mentioned. 

References:

Please ensure that journal name abbreviations match those found in the National Center for Biotechnology Information (NCBI) databases (http://www.ncbi.nlm.nih.gov/nlmcatalog/journals), and are appropriately formatted and capitalised. E.g. Ref 7 - Int J Gynaecol Obstet

Please also see https://journals.plos.org/plosmedicine/s/submission-guidelines#loc-references for further details on reference formatting. 

Comments from Reviewers:

Reviewer #1: Thanks authors for their effort to improvve the manuscript. The extra sensitivity analysis on seasonal effect is adequate and well reported. The statistical aspects of the paper are satisfactory. No further issues needing attention.

[LINK]

---

## [Editor Report · Decision Letter 4]

16 Mar 2022

Dear Dr Foo, 

On behalf of my colleagues and the Academic Editor, Prof. Lars Åke Persson, I am pleased to inform you that we have agreed to publish your manuscript "Prenatal influenza vaccination and allergic and autoimmune diseases in childhood: A longitudinal, population-based linked cohort study" (PMEDICINE-D-21-02861R4) in PLOS Medicine.

PRESS

Sincerely, 

Louise Gaynor-Brook, MBBS PhD 

PLOS Medicine